# The Transient Unloading Response of a Deep-Buried Single Fracture Tunnel Based on the Particle Flow Method

Xiqi Liu [1], Gang Wang [1,2], Zhijie Wen [3,*], Dongxing Wang [1,*], Leibo Song [2], Manqing Lin [4] and Hao Chen [2]



1   Key Laboratory of Geotechnical and Structural Engineering Safety of Hubei Province, School of Civil Engineering, Wuhan University, Wuhan 430070, China
2   Collaborative Innovation Center for Prevention and Control of Mountain Geological Hazards of Zhejiang Province, Shaoxing University, Shaoxing 312000, China
3   State Key Laboratory of Mining Disaster Prevention and Control, Shandong University of Science and Technology, Ministry of Education, Qingdao 266590, China
4   School of Resources and Safety Engineering, Wuhan Institute of Technology, Wuhan 430070, China
*   Correspondence: 2021000077@usx.edu.cn (Z.W.); dongxing-wang@whu.edu.cn (D.W.);
    Tel.: +86-18624326312 (Z.W.); +86-15271935123 (D.W.)

**Abstract:** Particle flow numerical simulation was used to reproduce the transient unloading process of a deep-buried single fracture tunnel. The influence of fracture characteristics on the transient unloading effect was analyzed from the aspects of stress state, deformation characteristics, fracture propagation, and energy conversion. The results shows that the surrounding rock stress field of the deep-buried tunnel is divided into four areas: weak stress area I, strong stress area II, stress adjustment area III, and initial stress area IV. The fracture has an important impact on the stress adjustment process of transient unloading of the deep-buried tunnel, and the stress concentration area will be transferred from the bottom corner of the chamber and the vault to the fracture tip. With the increase in the fracture length, the distance from the stress concentration area at the fracture tip to the free surface gradually increases, and the damage area of the surrounding rock gradually migrates to the deep area of the rock mass. At this time, the release amount of strain energy gradually decreases and tends to be stable, while the dissipation energy shows a near 'U' shape change trend of decreasing first and then increasing. Under different fracture angles, the number of mesocracks is significantly different. Among them, the number of mesocracks in the 60° and 30° fractured surrounding rocks is greater followed by the 0° fractured surrounding rock, and the number of mesocracks in the 45° and 90° fractured surrounding rocks is relatively less. In addition, the proportion of compression-shear cracks shows a change trend of increasing first and then decreasing with the increase in the fracture angle, and it reaches the maximum value in the 45° fractured surrounding rock.

**Keywords:** deep-buried tunnel; fractured rock mass; transient unloading; crack propagation; energy conversion

## 1. Introduction

Deep rock mass often contains different degrees of structural planes such as fractures, joints, and even large faults. Such defects largely determine the strength of the rock mass. When subjected to external loads such as high ground stress and unloading, the fracture damage of the surrounding rock is aggravated, and the defects of the structural plane expand and penetrate to the instability and the failure of the rock mass [1–3]. In the process of excavation and unloading, the existence of structural planes will greatly reduce the service life of rock mass engineering and correspondingly increase the economic costs of construction and safety protection. Therefore, it is of great significance to study the transient unloading response of deep-buried fracture tunnels for promoting socially sustainable development and improving the socio-economic value of engineering.

The deformation of rock mass under the action of external force and the failure and instability after reaching the bearing strength are the results of fracture initiation, propagation, condensation, and penetration in rock driven by energy [4–6]. Therefore, it is of great significance to analyze the transient unloading effect of a deep buried fractured tunnel from the perspectives of energy conversion and fracture evolution. A large number of scholars have carried out a series of studies on the excavation and unloading failure mechanism and mechanical characteristics of the surrounding rock of roadways under in situ stress environments and have obtained many research results. For example, Martin et al. [7] studied the failure mode of a circular tunnel after excavation transient unloading, found that the failure model was mainly "V" type, explored its failure mechanism through a model test, and qualitatively explained the reasons for different types of failure. Torano et al. [8] found that after the transient unloading of deep rock mass, due to the emergence of a new free surface, there will be greater tensile stress in the surrounding rock, resulting in the tensile failure and cracking of the surrounding rock. Li et al. [9] and Cao et al. [10] studied the loosening and deformation response of surrounding rock in the process of tunnel excavation and unloading, put forward the strength-influencing factors that induce the vibration of surrounding rock, and analyzed the influence of unloading wave on the disturbance effect of surrounding rock in a tunnel. Huang et al. [11] analyzed the deformation and failure of rock mass excavation according to the energy conversion law during unloading and explored the transformation process of strain energy. It was concluded that the unloading rate would affect the transformation of strain energy in rock and then determine the failure mode of rock, and the dissipation energy during excavation was also affected by the unloading rate. However, considering that various forms and different types of fractures are inevitably distributed in deep rock mass, some scholars have analyzed the fracture failure characteristics of fractured rock unloading from the perspective of fracture characteristics. These studies have certain reference significance for guiding practical engineering applications. For example, Huang et al. [12] found that the failure mode of fractured rock mass or rock bridge is determined by the inclination angle of prefabricated cracks, initial stress state, and unloading conditions. Based on the triaxial unloading test, Huang et al. [13] found that the confining pressure and unloading rate have significant effects on the rock failure mode and strain energy conversion, and unloading normal stress will significantly reduce the shear strength of rock. He et al. [14] summarized the dynamic damage process and characteristics of limestone under dynamic unloading based on the true triaxial single-side dynamic unloading test. The research of Hogan et al. [15] showed that there is a size effect on the unloading failure degree of granite.

In recent years, the PFC discrete element analysis method has been widely used in the field of geotechnical engineering. It constructs a particle numerical model from a mesoscopic point of view, reflecting the continuous nonlinear stress–strain relationship of materials and the spatiotemporal evolution process of fracture initiation, propagation, and penetration through the failure of inter-particle contact [16,17]. Lee et al. [18] established a variety of fractured rock mass models with different types and different inclination angles through different contact bond models, used a particle flow program to simulate uniaxial and triaxial compression tests of rocks, and studied the influence of fracture shape and distribution on the initiation, propagation, and penetration process of mesocracks in rocks. Fan et al. [19] first obtained macroscopic parameters through laboratory tests and calibrated the mesoscopic parameters of the particle flow model and then established different types of multi-fracture particle models to study the impact of parallel fractures and cross-fractures on the deformation and failure of unloaded rock. Wang et al. [20], based on PFC2D, established a variety of defective coal rock samples taking into account the fractures with different inclination angles and holes at different positions and discussed the stress–strain characteristics and fracture evolution laws of coal rock with different holes and fractures. In previous studies, there were few studies on rock mass with different fracture distributions, especially the impact of different fracture distributions on the transient unloading effect of deep-buried tunnels.

In this paper, particle flow numerical simulation is used to reproduce the transient unloading process of a deep buried single fracture tunnel, and the influence of fracture characteristics on the transient unloading effect of deep rock mass is analyzed from multiple perspectives such as stress state, deformation characteristics, fracture propagation, and energy conversion. The research results have guiding significance for the understanding and evaluation of surrounding rock failure mechanisms under the excavation and unloading effect of similar underground powerhouse caverns.

## 2. Particle Flow Model of the Deep-Buried Single Fracture Tunnel

### 2.1. Contact Constitutive Model

Particle flow numerical simulation is a discrete analysis element method. It can construct any rock morphology into a particle aggregate, simulate the macroscopic mechanical properties of rock through the interaction between particles, and simulate the generation of cracks in rock materials through the fracture of bonding between particles [21,22]. In this paper, based on the flat-joint contact model, the biggest difference between the flat-joint contact model and the parallel bond model and the contact bond model is that it can suppress rotation after particle bond failure [23]. As shown in Figure 1, the particle shapes are constructed into a polygon and "occluded" with each other. When particle bond failure occurs, due to the mutual "occluded" effect between the particles, the particle unit cannot rotate freely and can only slide or fall off as a whole, which is closer to the internal microstructure of the actual rock material.

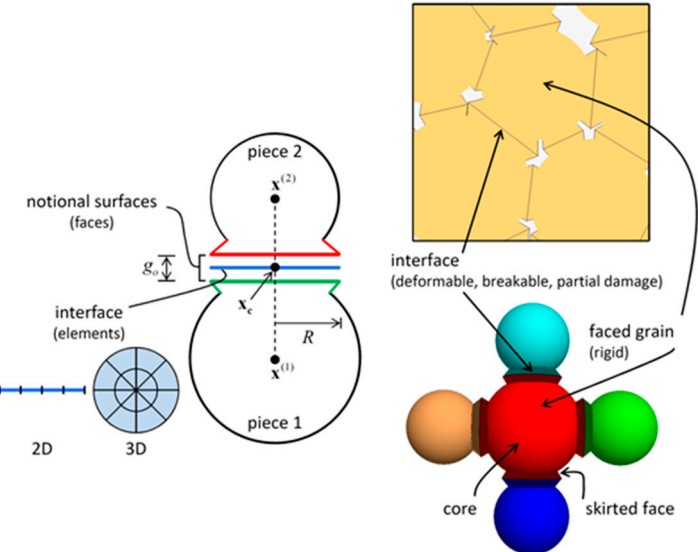

**Figure 1.** Flat-joint contact model.

### 2.2. Particle Flow Model

The structure boundary of the tunnel is a semi-infinite body boundary, and the particle flow program cannot carry out numerical simulation for such a large range of rock material models, and tunnel excavation has little impact on the rock mass far away. Therefore, in numerical simulation, a limited calculation range is generally selected to establish a model, as long as the calculation range is larger than the influence range of tunnel excavation. A lot of engineering experience shows that the influence range of tunnel structure is generally within 3–5 times the tunnel characteristic size from the tunnel center [24]. Taking the traffic tunnel of Shuangjiangkou Hydropower Station as the research background, in order to simulate the transient unloading process of the tunnel, the calculation range of the model is set to be 26 m × 26 m, the height of the chamber side wall is 2 m, the bottom width is 4 m, and the arch height is 2 m. The chamber is located in the center of the model, and the tunnel center coincides with the center of the calculation model. The numerical model establishment process is mainly divided into two steps. Firstly, the

particle expansion method is used to generate the entire particle flow model, with a number of 34,880 particles generated. After the particle flow model is gravity balanced, the velocity field and displacement field of the calculation model are cleared to zero. Then, a tunnel excavation model is established, and transient unloading is simulated by deleting particles within the tunnel range. The numerical model established is shown in Figure 2. The model calculates until the average ratio is less than or equal to the equilibrium tolerance. The average ratio is the ratio of the average value of the unbalanced force magnitude over all bodies to the average value of the sum of the magnitudes of the contact forces, body forces, and applied forces over all the bodies.

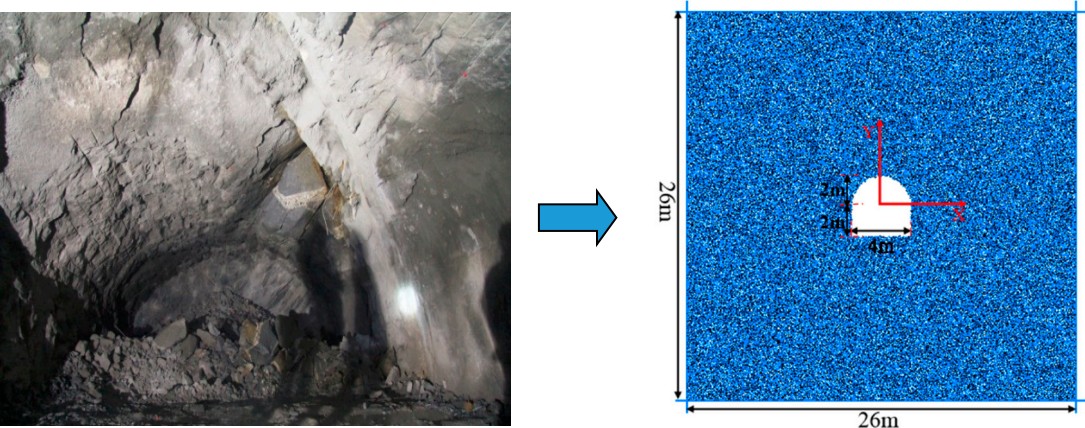

**Figure 2.** PFC model of tunnel rock mass excavation.

In this paper, the research is carried out from the aspects of fracture length, angle, etc., and the stress distribution state, deformation law, and failure characteristics of the fractured surrounding rock during the transient unloading process of the deep-buried cavern are analyzed. The difference between vertical and horizontal ground stresses in the surrounding rock of the diversion tunnel of Shuangjiangkou Hydropower Station is small, and the average principal stress value is about 20 MPa. Therefore, in the numerical simulation, the ground stress effect is simulated by applying 20 MPa confining pressure on the upper and lower boundary walls and the left and right boundary walls. The constant confining pressure is achieved by the servo system. The wall servo provides the ability to control the translational velocity of select walls using a servo-mechanism in order to apply or maintain a target confining stress. The wall velocity is decomposed into two components, namely the signed magnitude of the velocity and the unit vector in the direction of the target force.

The numerical model of complete rock material with the flat-joint contact model as a constitutive relation [25,26] can more truly reflect the macroscopic mechanical properties of rock materials and the numerical model of fractured rock is established by setting corresponding fracture elements. Considering the non-penetrating open fractures in the actual fractured rock, the relevant mesoscopic parameters in the smooth-joint contact model are selected, including the contact element normal stiffness $k_{nj}$, the contact element tangential stiffness $k_{sj}$, the friction coefficient $\mu_j$, and the contact element internal friction angle $\phi_j$. The remaining mesoscopic parameters can be taken as default values. According to the research of Huang et al. [13], the mesoscopic parameters of the contact model are finally obtained as shown in Tables 1 and 2.

**Table 1.** Calibration results of mesoscopic parameters of the tunnel rock model.

| $\lambda$ | $N$ | $\rho$/(kg/m³) | $R_{max}/R_{min}$ | $R_{min}$/mm | $E_c$/GPa | $k_n/k_s$ | $\mu$ | $\sigma_c$/MPa | $c_c$/MPa | $\phi$/° |
|---|---|---|---|---|---|---|---|---|---|---|
| 1 | 4 | 2500 | 1.66 | 45 | 10 | 2.38 | 0.55 | 2.5 | 10 | 21 |

**Table 2.** Mesoscopic parameters of the joint contact model.

| $k_{nj}$/GPa | $k_{sj}$/GPa | $\mu_j$ | $\sigma_{cj}$/MPa | $c$/MPa | $\phi$/° |
|---|---|---|---|---|---|
| 1 | 1 | 0.3 | 0 | 0 | 30 |

*2.3. Working Condition Design*

In order to study the influence of the fracture length and inclination angle on the transient unloading effect of the tunnel, considering the horizontal penetrating fracture which has a great impact on the failure characteristics of the surrounding rock, according to the different fracture lengths *L*, it can be divided into six working conditions, namely 5 m, 6 m, 7 m, 8 m, 9 m, and 10 m. The fracture center coincides with the geometric center of the model and runs through the tunnel and the specific setting is shown in Figure 3a. Based on the consideration of tunnel structure and calculation model size, a 6 m-long penetrating fracture is selected, and the fracture center and the geometric center of the tunnel structure are merged and run through the tunnel. The influence of different fracture angles on the transient unloading effect of the deep-buried tunnel is explored by changing the angle *α* between the fracture surface and the horizontal *X*-axis. The fracture angle *α* is set to 0°, 30°, 45°, 60° and 90° in turn, as shown in Figure 3b. Among them, the fractures are horizontally distributed in the initial state, and the fractures within the scope of the tunnel excavation face are also destroyed transiently during transient unloading. The excavation process under each working condition is calculated to be 50,000 steps.

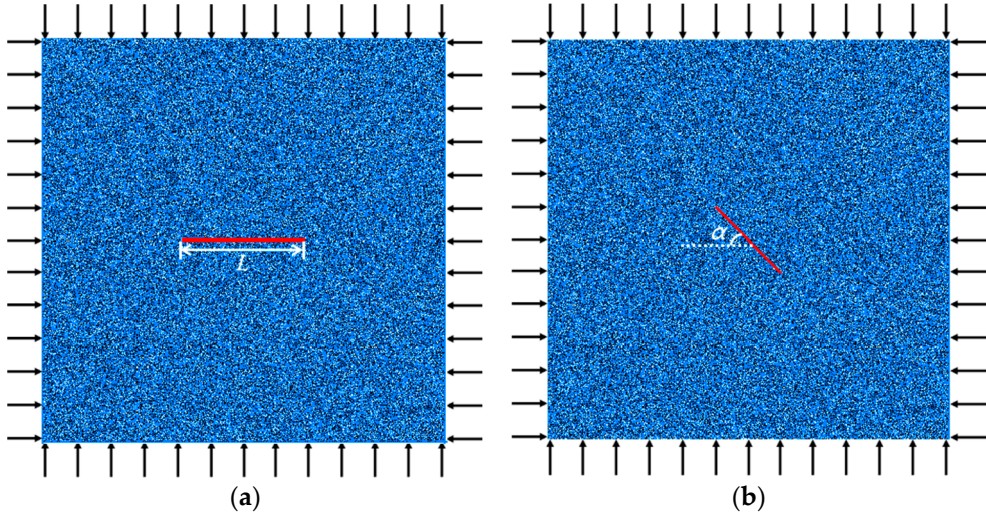

(a)         (b)

**Figure 3.** Working condition design diagram. (**a**) Fracture length. (**b**) Fracture inclination angle.

**3. Numerical Calculation Results and Analysis**

*3.1. The Influence of Fracture Length on the Transient Unloading Effect of the Deep-Buried Tunnel*

3.1.1. Stress Characteristic Analysis

In the process of tunnel excavation and unloading, the surrounding rock can be roughly divided into four parts according to the stress distribution state (Figure 4b): weak stress zone I, strong stress zone II, stress adjustment zone III, and initial stress zone IV. When the tunnel is excavated and unloaded, the radial stress and hoop stress are suddenly released. In weak stress zone I closest to the tunnel contour, the surrounding rock is deformed greatly, the structure is destroyed, and the stress is released, leading to a sharp decrease in the bearing capacity. After that, due to the overall effect of the surrounding rock, strong stress zone II can still bear a large load although it is deformed to some extent. In stress adjustment zone III, the radial stress increases slowly, while the hoop stress begins to increase gradually. The surrounding rock tends to be stable, and the deformation is small. The final initial stress zone IV is the farthest from the tunnel structure and is slightly

affected by excavation and unloading. The stress state is similar to the initial condition. The change of the hoop stress in the surrounding rock is the result of its self-adjustment. The strong stress zone has a strong bearing capacity, preventing further deformation and failure of the surrounding rock.

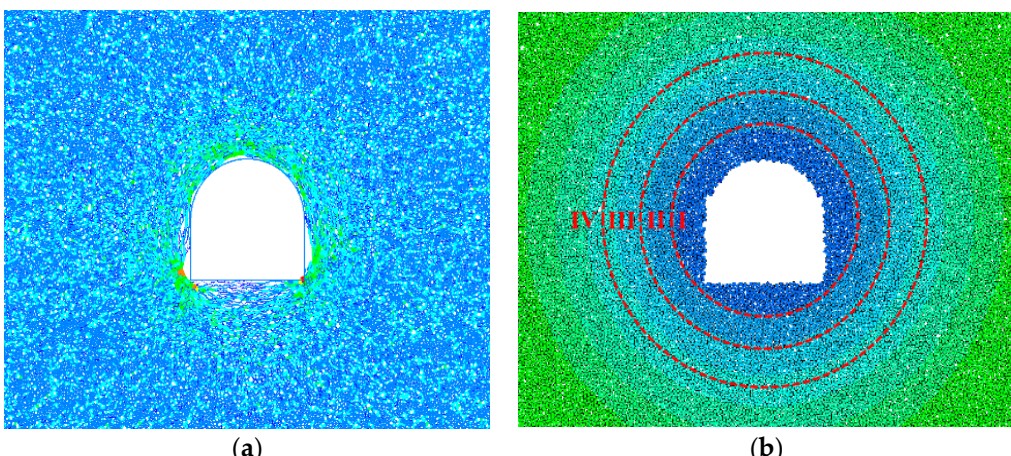

| (a) | (b) |

**Figure 4.** Stress distribution and partition map of excavation and unloading of the complete surrounding rock. (**a**) Stress distribution. (**b**) Stress partition.

Figure 5 shows the stress distribution and stress partition map under different fracture lengths after 50,000 steps of the excavation cycle. Under the condition of the complete surrounding rock, there is an obvious stress concentration at the left and right bottom corners, arch shoulders, and vaults of the tunnel, and the stress distribution at the tunnel excavation boundary is dense. The maximum stress appears at the bottom corners of the tunnel, and the tunnel-surrounding rock is severely damaged. In the fractured rock mass, the stress is not concentrated at the excavation surrounding the rock boundary but mainly distributed at the initial fracture tip. With the increase in the fracture length, the distance between the part with a large stress concentration and the tunnel boundary gradually increases, the stress distribution around the tunnel is gradually sparse, and the stress accumulation at the left and right bottom corners and the vault is continuously reduced. It can be seen that the fractures have an important impact on the stress adjustment process of the transient unloading of the deep-buried tunnel, and the existence of the fractures changes the stress adjustment areas after tunnel excavation.

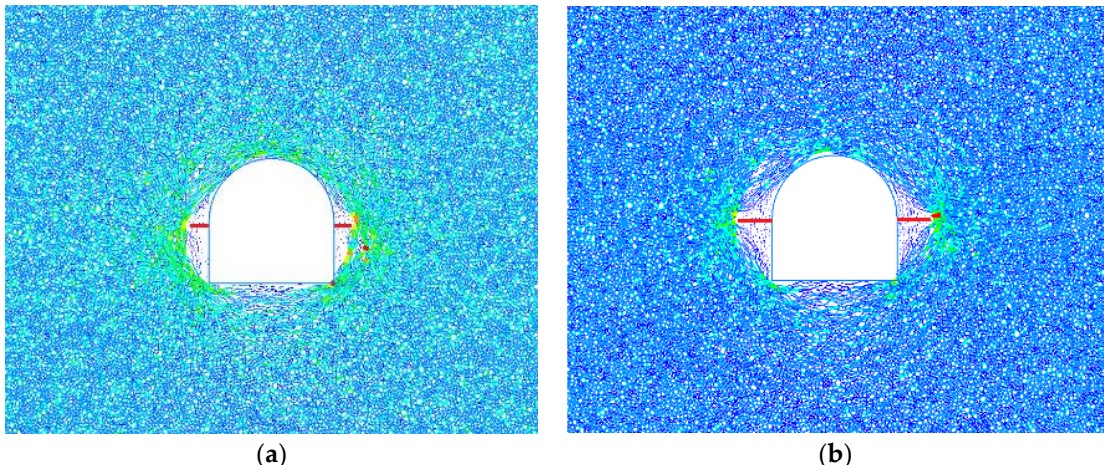

| (a) | (b) |

**Figure 5.** *Cont.*

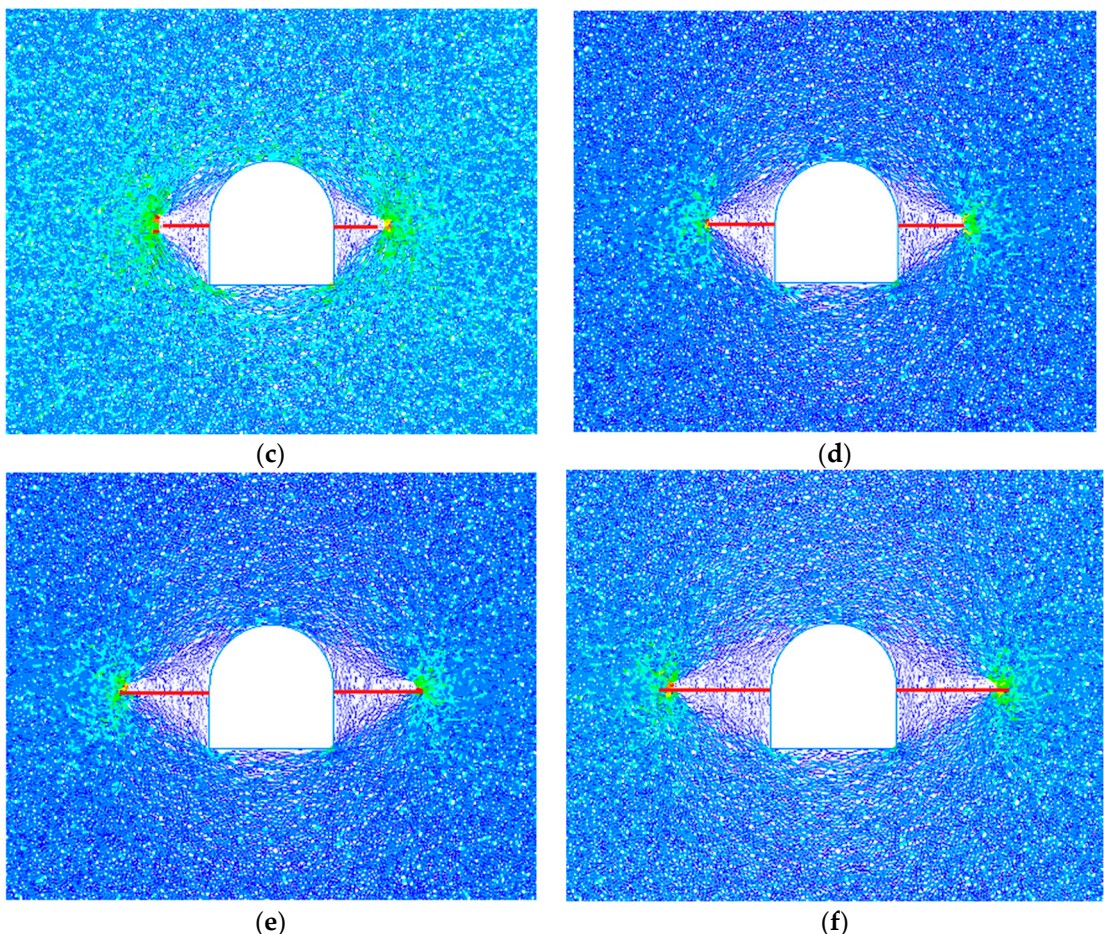

**Figure 5.** Stress distribution under different fracture lengths. (**a**) 5 m; (**b**) 6 m; (**c**) 7 m; (**d**) 8 m; (**e**) 9 m; (**f**) 10 m.

### 3.1.2. Deformation Evolution Law

In the process of tunnel excavation, the displacement of surrounding rock directly affects the fragmentation degree of the tunnel structure and is also an important indicator to measure the stability of the surrounding rock. The displacement changes in the surrounding rock under transient unloading with different fracture lengths are almost the same. Taking the numerical model of a 6 m fracture length as an example, the displacement direction in the surrounding rock is shown in Figure 6. It can be seen that the self-adjustment of fractured surrounding rock is mainly reflected in three aspects: ① the surrounding rock at the vault shows a sinking trend, but the surrounding rocks at the arch shoulder tend to squeeze toward the center while moving in opposite directions, forming the "arching" effect, thus improving the bearing capacity of the vault. ② The surrounding rocks at the two sides and the left and right bottom corners show the trend of opposite movement, so that the stress of the whole tunnel structure tends to be balanced, and large deformation and damage can be avoided locally. ③ The surrounding rocks at the bottom plate of the arch are squeezing each other, showing an upward trend.

In order to further explore the deformation law of the tunnel structure, six measuring units are set at the key points of the tunnel model, such as the left and right bottom corners, the left and right arch shoulders, and the vault and the center of the bottom plate to monitor the surrounding rock displacement, as shown in Figures 7 and 8, and they show the particle displacement at each key point. It can be seen from the figures that the displacement of the complete surrounding rock at each key point is greater than that of the fractured surrounding rock, and the displacement of the complete surrounding rock at the No. 5 key point exceeds 4.5 times the maximum displacement of the fractured surrounding rock.

This is because the existence of horizontal penetrating fractures increases the damage and deformation range of the tunnel-surrounding rock and relatively reduces the displacement of the surrounding rock around the tunnel contour. Under different fracture lengths, the displacement at the same key point decreases and tends to be stable as the fracture length increases. Under the same fracture length, the displacement at each key point is significantly different. The difference in the surrounding rock displacement between the No. 2 key point and the No. 4 key point is small, while the difference in the surrounding rock displacement between the No. 1 key point and the No. 5 key point is relatively large, and the displacement of the No. 1 key point is greater than that of the No. 5 key point. Among all key points, the displacement at the No. 3 key point is always the largest, and the displacement at the No. 5 key point is always the smallest. The displacement of the surrounding rock at the No. 3 key point is greater than that at the No. 6 key point, but the difference decreases with the increase in the fracture length. The difference between the two decreases by nearly 43.3% from the 5 m fracture to the 10 m fracture. It is indicated that the displacement of the vault is always greater than that of the bottom plate in the surrounding rock with horizontal penetrating fractures of different lengths, but the difference decreases with the fracture length.

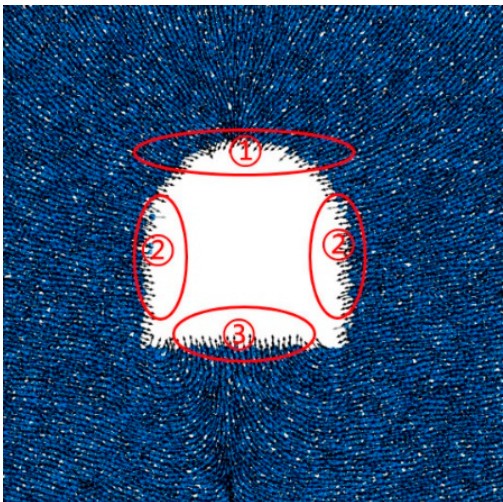

**Figure 6.** Particle displacement field of the fractured surrounding rock.

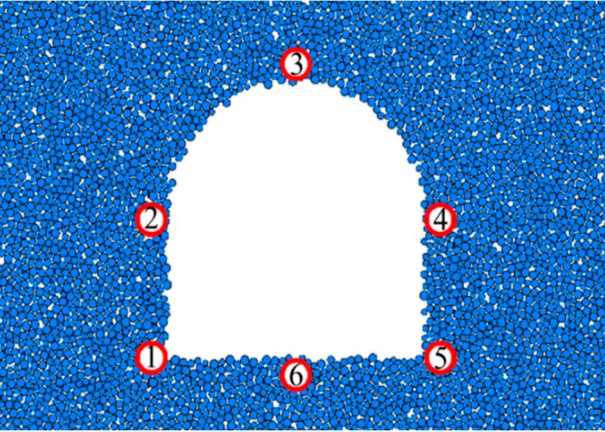

**Figure 7.** Key point setting.

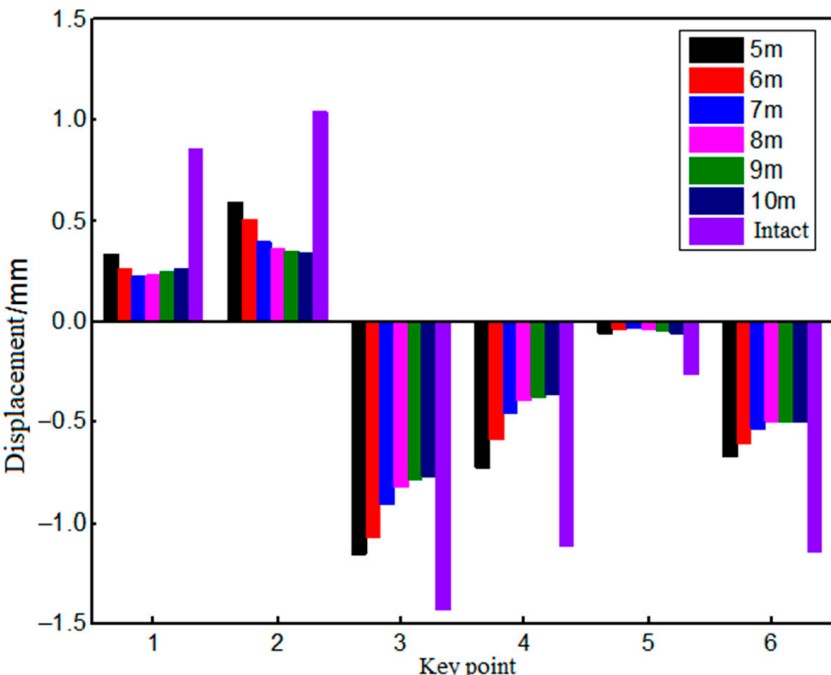

**Figure 8.** Displacement at key points in tunnels with different fracture lengths.

### 3.1.3. Fracture Propagation Law

Figure 9 shows the distribution of mesocracks in the surrounding rock of the tunnel under different fracture lengths. In the figure, the black line represents the mesoscopic compression shear crack, the green line represents the mesoscopic tensile crack, and the mesocracks are mainly distributed around the tunnel structure. The initiation, expansion, accumulation, and intersection of cracks cause the deterioration of the surrounding rock structure and the weakening of mechanical properties. In the process of stress redistribution, due to the normal unloading of the excavation face, the tangential stress concentration of the surrounding rock is caused, and the compression-shear crack propagation occurs in the direction parallel to the excavation face, leading to the compression-shear failure of the surrounding rock. After the transient unloading of the deep-buried tunnel, the surrounding rock is mainly subjected to compression-shear failure, corresponding to the dominant black shear cracks in the figure. The shear cracks of the complete surrounding rock are concentrated near the bottom plate and two sides, while a small number of tensile cracks are distributed at the left and right bottom corners. The distribution of mesocracks in the surrounding rock varies greatly under different fracture lengths. When the fracture length is small, the distribution of mesocracks is relatively dense and mainly concentrated in the bottom plate, two sides, and arch shoulders. When the fracture length is large, the fracture distribution is sparse, but the extension range is large. There is a small amount of fracture accumulation near the bottom plate and vault. The fractures are scattered on the two sides and extend to the initial fracture tip. It can be seen that with the increase in the fracture length, the distance from the stress concentration point at the fracture tip to the free surface gradually increases, and the damage range of the surrounding rock continues to expand. The distribution of mesocracks on the two sides of the tunnel gradually becomes sparse, the extended area gradually becomes larger, and the damaged area of surrounding rock gradually migrates to the deep area.

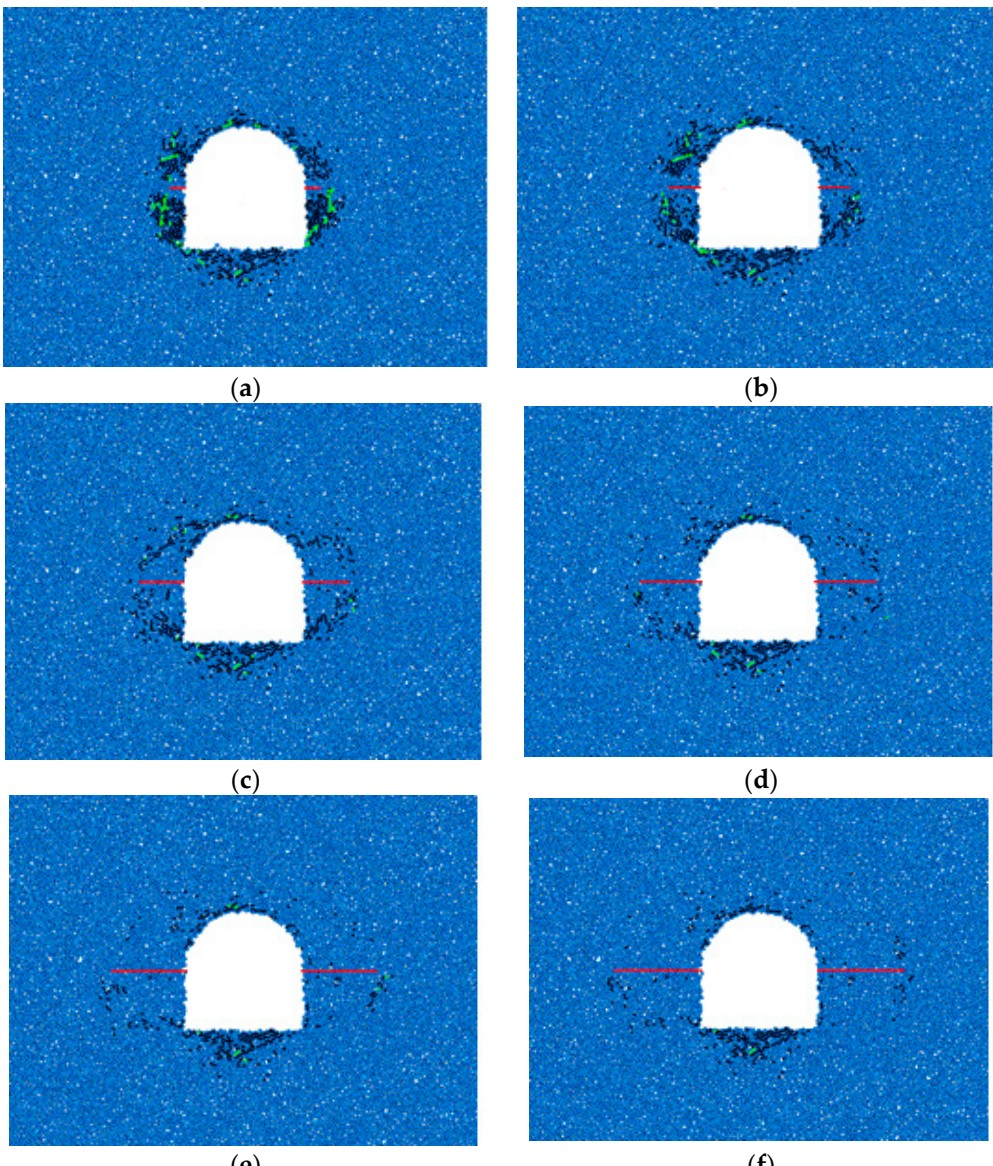

**Figure 9.** Distribution of mesocracks in the tunnel-surrounding rock under different fracture lengths. (**a**) 5 m; (**b**) 6 m; (**c**) 7 m; (**d**) 8 m; (**e**) 9 m; (**f**) 10 m.

When the surrounding rock is damaged and destroyed, the tensile and shear cracks represent different failure mechanisms. The statistical data on the number of mesocracks are shown in Table 3. According to the data in the table, the number of fractures and the proportion of each fracture under different fracture lengths are compared and analyzed, as shown in Figure 10. Under different fracture lengths, the total number of mesocracks and the proportion of tensile and shear cracks are different. When the fracture length is small (5 m and 6 m), the total number of mesocracks in the fractured surrounding rock is more than that in the complete surrounding rock, while the total number of mesocracks in the surrounding rock under other fracture lengths is lower than that in the complete surrounding rock. When the fracture length is 5 m, the proportion of shear cracks is 88.4%, which is lower than 91.5% of shear cracks in the complete surrounding rock, and the proportions of shear cracks under other fracture lengths are higher than those in the complete surrounding rock. On the whole, with the increase in the fracture length, the total number of mesocracks shows a monotonic decreasing trend, from 1048 to 403, with a decrease of up to 62%. The proportion of shear cracks increases monotonically with

the fracture length, from the lowest value of 88.4% to the highest value of 98.5%, with an increase of 11.4%.

**Table 3.** Statistical results of crack number.

| Crack Length/m | Total Number of Cracks | Compression-Shear Crack | | Tensile Crack | |
|---|---|---|---|---|---|
| | | Number of Cracks | Proportion | Number of Cracks | Proportion |
| 5 | 1048 | 926 | 88.4% | 122 | 11.6% |
| 6 | 900 | 839 | 93.2% | 61 | 6.8% |
| 7 | 683 | 658 | 96.3% | 25 | 3.7% |
| 8 | 527 | 509 | 96.6% | 18 | 3.4% |
| 9 | 423 | 409 | 96.7% | 14 | 3.3% |
| 10 | 403 | 397 | 98.5% | 6 | 1.5% |
| Intact | 779 | 713 | 91.5% | 66 | 8.5% |

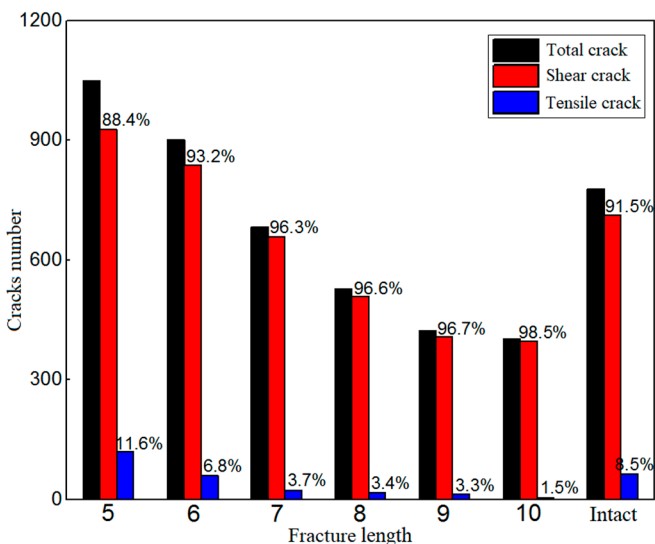

**Figure 10.** The variation diagram of fracture with fracture length.

3.1.4. Energy Conversion Law

Energy conversion is the essential feature of various physical changes. In the process of transient unloading of deep-buried tunnel, the damage and deformation of the surrounding rock is also the result of energy release and transformation. The PFC discrete element analysis method can track the change in energy storage and release during the simulation process. Through monitoring the total energy and elastic strain energy in the whole model system, the released amount of strain energy caused by tunnel transient unloading and the total amount of dissipation energy in the whole process can be calculated. The energy conversion law is shown in Figure 11. The release amount of strain energy and the total amount of dissipation energy during the transient unloading of surrounding rock with different fracture lengths are less than the energy value of the complete surrounding rock. The release amount of strain energy is the largest when the fracture length is 5 m, which can reach 91% of the integral surrounding rock. When the fracture is 10 m-long, the total amount of dissipation energy is largest, accounting for 90% of the complete surrounding rock. The existence of fractures destroys the integrity of rock mass and releases some elastic strain energy to a certain extent. Tunnel excavation has a certain influence range, and with the increase in the fracture length, the influence will gradually weaken and tend to disappear, namely, the energy released is closely related to the tunnel size. The released strain energy decreases first and then tends to be stable. With the increase in the fracture length, the total amount of dissipation energy decreases first and then increases. The

damaged area also migrates from the surrounding area of the tunnel to the deep part of the rock mass.

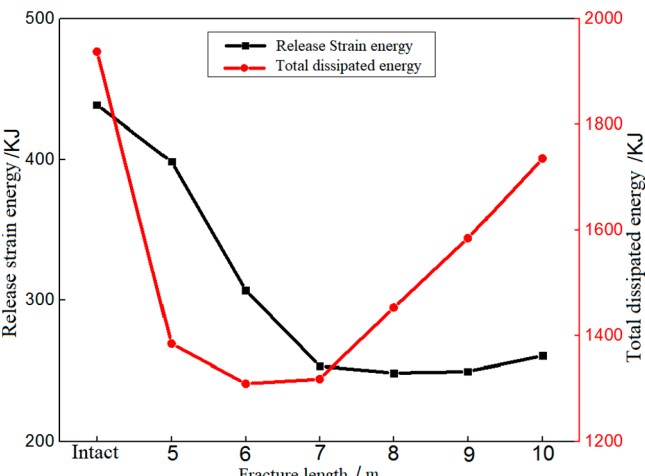

**Figure 11.** Energy comparison diagram under different fracture lengths.

*3.2. The Influence of the Fracture Inclination Angle on the Transient Unloading Effect of the Deep-Buried Tunnel*

3.2.1. Stress Characteristic Analysis

In the particle flow model, the contact force chain between particles can be approximately used to characterize the stress distribution in the surrounding rock. Figure 12 shows the stress distribution in the surrounding rock after 50,000 steps of the tunnel excavation cycle under different fracture angles. It can be seen from the figure that the stress distribution in the surrounding rock changes synchronously with the change in the fracture angle. In the complete surrounding rock, the stress is densely distributed around the tunnel structure, and the stress is mainly concentrated at the left and right bottom corners and the vault, and the stress value at the bottom corner is the largest. In the fractured surrounding rock, the stress is mainly concentrated at the fracture tip. In addition, stress accumulation will also occur at the left and right bottom corners and the vault of the tunnel. In the 0° fractured surrounding rock, the stress accumulated at the fracture tip extends to the periphery of the tunnel structure in an arc shape, and a certain stress concentration is generated at the left and right bottom corners and the vault. The stress near the two sides is relatively small, and the final stress shows an approximate symmetric annular distribution. In the 30°, 45°, and 60° fractured surrounding rocks, due to the existence of oblique fractures, the stress distribution is uneven. The distance from the fracture tip to the tunnel excavation contour is different, and the stress concentration at the fracture tip is also different. The stress concentration at the tip far from the free surface is more intensive, while the stress distribution at the tip near the free surface is relatively sparse. Under the action of oblique fractures, the stress chain around the tunnel structure is relatively sparse, the stress that the surrounding rock can bear is reduced, and the stress concentration mainly occurs at the bottom corner and the arch shoulder far away from the fracture tip. In the 90° surrounding rock, a large number of stress chains accumulate at the upper tip of the fracture and extend to the arch shoulder of the tunnel. The stress concentration occurs at the lower tip of the fracture and at the left and right bottom corners of the tunnel, and the three are connected in an arc shape. The stress chains near the two sides of the tunnel are relatively dense, and the stress is approximately symmetrically distributed along the direction of the fracture surface.

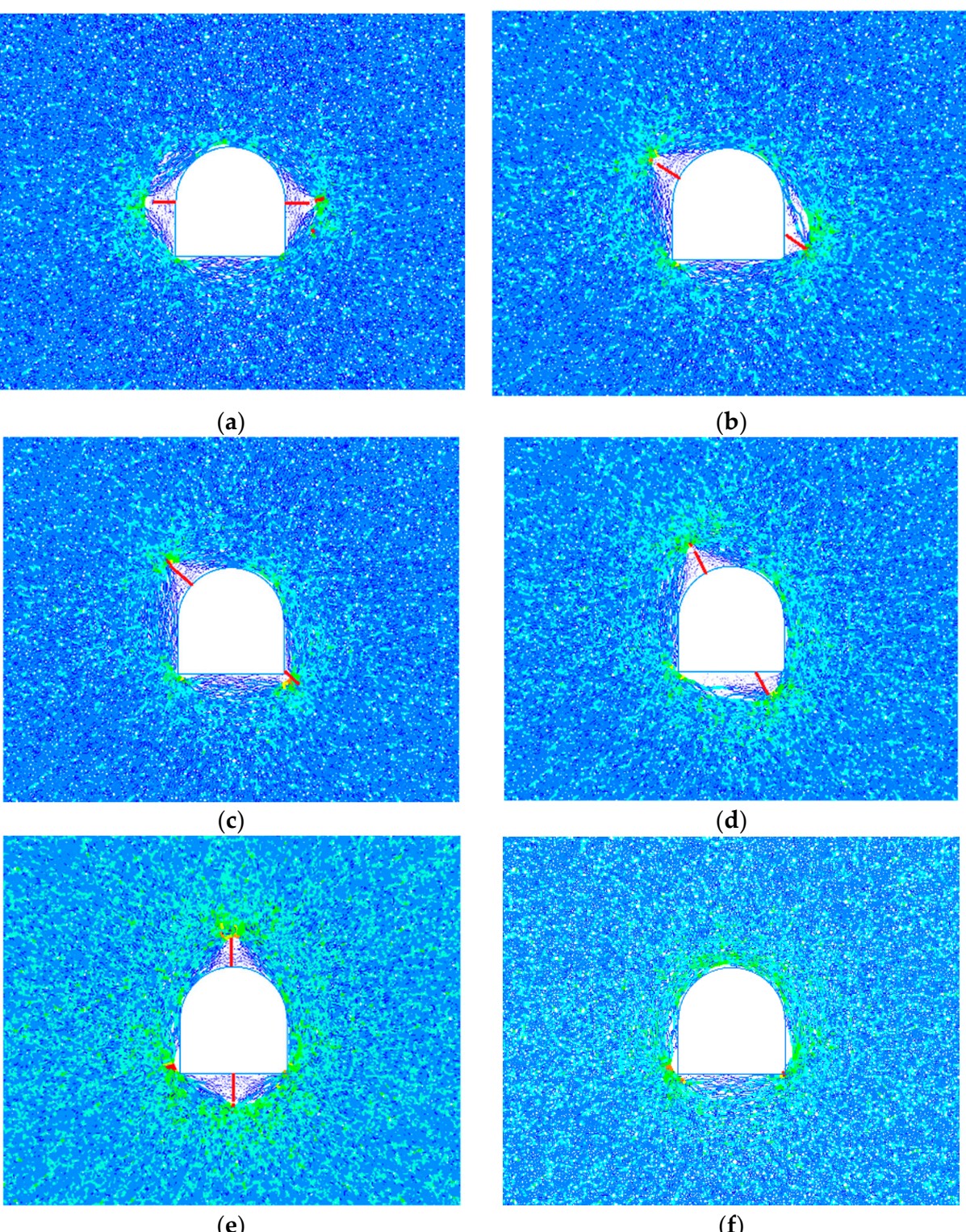

**Figure 12.** Stress distribution of the surrounding rock under different fracture angles. (**a**) 0°; (**b**) 30°; (**c**) 45°; (**d**) 60°; (**e**) 90°; (**f**) Intact rock.

### 3.2.2. Deformation Evolution Law

In the particle flow program, the displacement distribution of model particles can approximately represent the displacement field of the tunnel-surrounding rock [27,28]. The displacements of surrounding rocks under different fracture angles are shown in Figure 13. The displacement field of the tunnel-surrounding rock varies greatly with different fracture

angles. Among them, the distribution of the displacement field in the complete surrounding rock is relatively symmetrical, and the displacement in the small range of the two sides of the tunnel is relatively large, while the displacement in the sector area under the bottom plate is relatively small. The displacement field in the 0° fractured surrounding rock is uniformly distributed, and larger displacements occur only in the tunnel vault and the small area of side walls. In the 30° fractured surrounding rock, there is a large displacement near the fracture tip, and the displacement at the upper left sector area is larger. The right-side wall is seriously deformed due to the large impact of the fracture, and the surrounding rock displacement is also large. In the 45° fractured surrounding rock, the surrounding rock on the left side of the tunnel structure has a large deformation, and the displacement in the crescent area on the upper left side is relatively large, while the displacement in the sector area under the bottom plate is small. In the 60° fractured surrounding rock, the displacement of the surrounding rock of the vault near the fracture tip is large, and the particles fall off. The displacement in the triangle area under the bottom plate is relatively small. In the 90° fractured surrounding rock, the deformation at the bottom corner of the tunnel is relatively serious and the displacement is large. The displacement in the arc area on the upper side of the vault is also large, while the displacement in the semicircle arc area under the bottom plate is relatively small. On the whole, the displacement of the tunnel-surrounding rock moves synchronously with the change of fracture inclination angles, but the displacement of surrounding rock under the bottom plate is always small.

The displacements of key points at different fracture angles are shown in Figure 14. It can be seen from the figure that the displacement of each key point of the fractured surrounding rock is less than that of the complete surrounding rock. At the No. 3 key point, the displacement of 30° fractured surrounding rock reaches the maximum value of 1.11 mm, which is 22.4% lower than that of the complete surrounding rock. Under the same fracture angle, the displacement of surrounding rock at each key point is quite different. The maximum displacement of 0°, 30°, 45°, and 60° fractured surrounding rocks occurs at the No. 3 point, while the maximum displacement of the 90° fractured surrounding rock occurs at the No. 1 point. This is because the 90° fracture vertically penetrates the vault, which has a great impact on the surrounding rock structure above the vault, resulting in insignificant deformation at the tunnel vault. With the increase in the fracture angle, the displacements of the surrounding rock at key points 1, 2, 4, and 5 show a trend of first decreasing and then increasing. The displacement of surrounding rock at the No. 3 key point increases first and then decreases. The displacement direction of the surrounding rock at the No. 6 key point is changed. The displacements in the 0° and 90° fractured surrounding rocks are downward, and the displacements in other angles are upward, indicating that the "arching" effect of the tunnel bottom plate is obvious under the influence of oblique fractures. At the No. 2, 4, and 6 key points, the displacement of the 0° fractured surrounding rock reaches the maximum. At the No. 1 and 5 key points, the displacement of the 90° fractured surrounding rock reaches the maximum. At the No. 3 key point, the maximum displacement of the surrounding rock occurs in the 30° fractured surrounding rock. At the No. 5 key point, the displacement of the surrounding rock is always the minimum.

### 3.2.3. Fracture Propagation Law

Figure 15 shows the distribution of mesocracks in the surrounding rock under different fracture angles. The dominant black cracks in the figure are compression-shear cracks, and the relatively few green cracks are tensile cracks. The tensile cracks are mostly distributed at the bottom corner, while the shear cracks are distributed in the surrounding area of the tunnel structure. Compared with the distribution pattern of mesocracks in the complete surrounding rock, the distribution range of mesocracks in the fractured surrounding rock with different angles is wider and far from the tunnel contour. Most of them are scattered along the tunnel periphery on both sides of the fracture, indicating that the damaged area of the surrounding rock moves synchronously with the fracture inclination angle. The

distribution pattern of mesocracks in the surrounding rock varies with different fracture angles. With the increase in the fracture angle, mesocracks in the surrounding rock near the two sides of the tunnel gradually become rare, while mesocracks in the surrounding rock under the bottom plate gradually increase. In the 30°, 45°, and 60° fractured surrounding rocks, the distribution of mesocracks is relatively dense, while in the 0° and 90° fractured surrounding rocks, the distribution of mesocracks is relatively scattered.

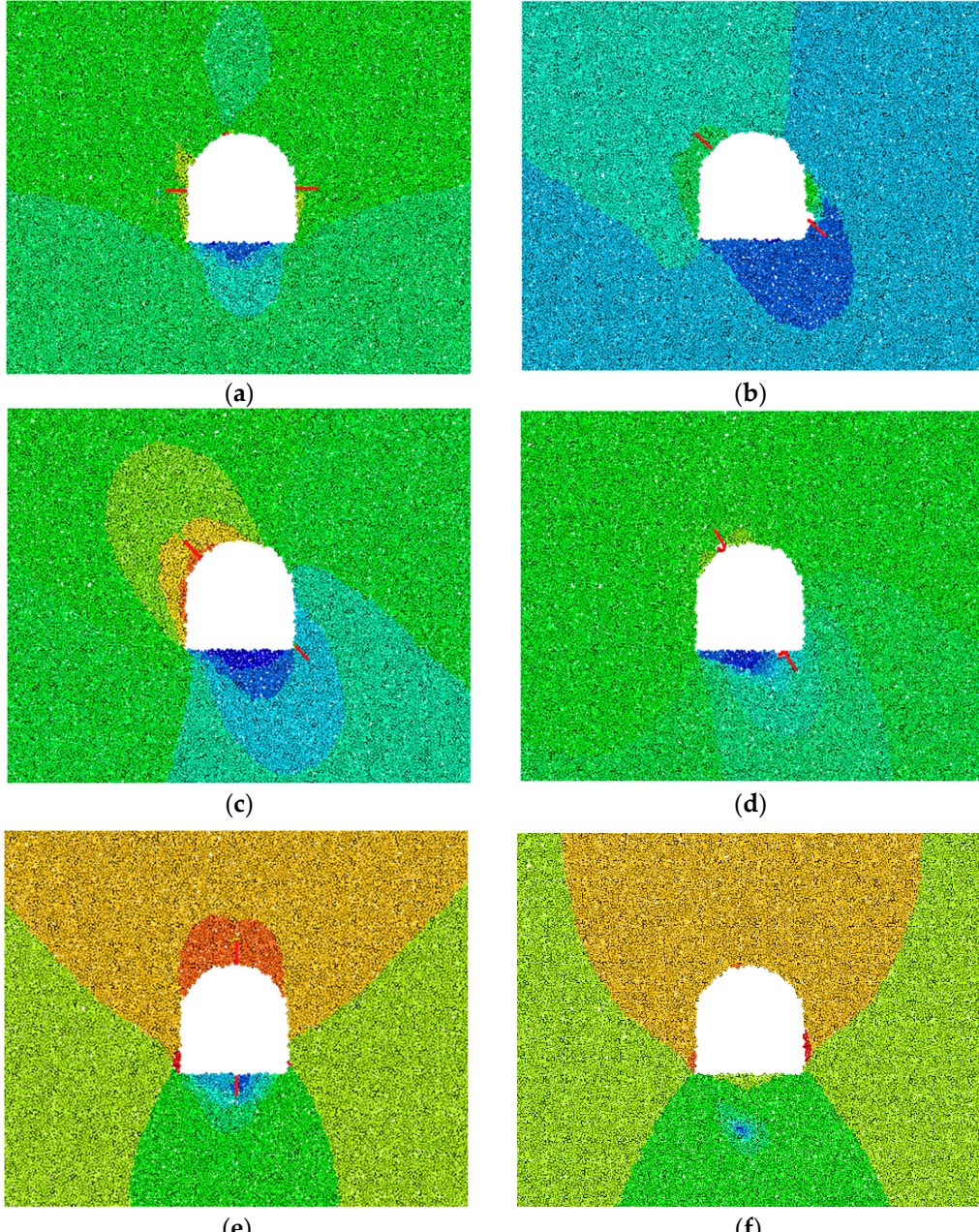

**Figure 13.** Displacement field distribution under different fracture angles. (**a**) 0°; (**b**) 30°; (**c**) 45°; (**d**) 60°; (**e**) 90°; (**f**) Intact rock.



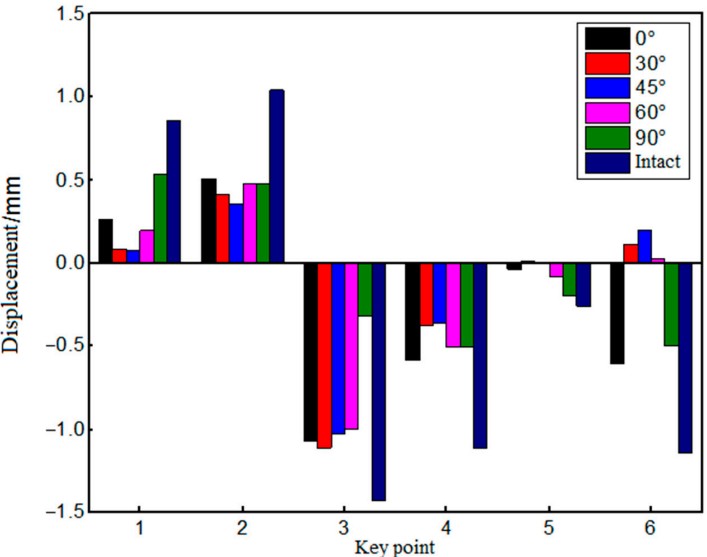

**Figure 14.** Displacement at key points in tunnels with different fracture inclination angles.

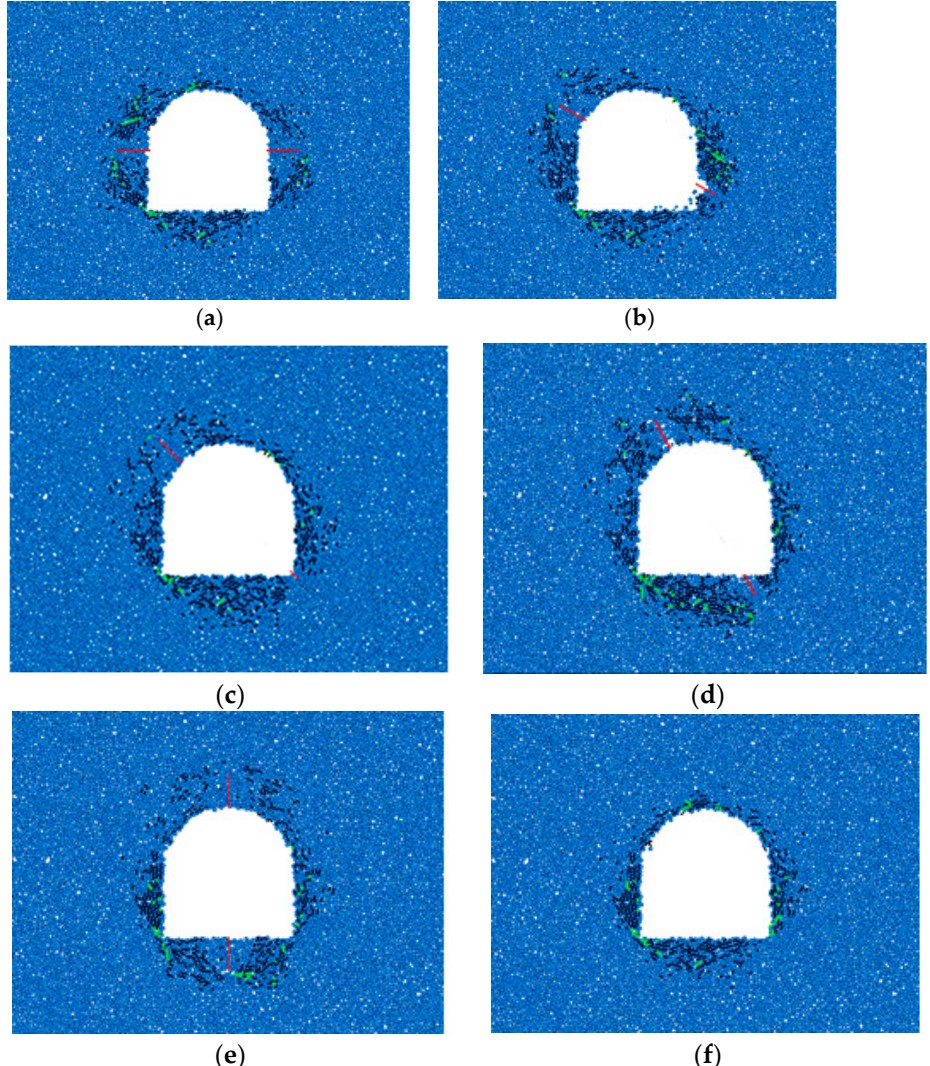

**Figure 15.** Distribution of mesocracks in the tunnel-surrounding rock under different fracture angles. (**a**) 0°; (**b**) 30°; (**c**) 45°; (**d**) 60°; (**e**) 90°; (**f**) Intact rock mass.

The statistics on the number of mesocracks in the damage process of the tunnel-surrounding rock under different fracture angles are shown in Table 4. According to the data in the table, the number of shear cracks is far more than that of tensile cracks, indicating that the surrounding rock is mostly damaged by compression and shear during the transient unloading process of the tunnel. The total number and proportion of fractures are compared and analyzed, as shown in Figure 16. The total number of mesocracks and the number of shear cracks in the surrounding rock under different fracture angles are more than those in the complete surrounding rock. The total number of mesocracks in the 60° fractured surrounding rock reaches the maximum value of 1066 mesocracks, which is nearly 37% higher than that in the complete surrounding rock and about 28% higher than the minimum value of 835 mesocracks in the 90° fractured surrounding rock. Under different fracture angles, the number of mesocracks varies greatly. The total number of mesocracks in the 60° and 30° fractured surrounding rocks is greater, followed by the 0° fractured surrounding rock, and the total number of mesocracks in the 45° and 90° fractured surrounding rocks is relatively less. The proportion of compression-shear cracks shows a trend of increasing first and then decreasing with the increase in the fracture angle. However, the number of mesocracks in the fractured surrounding rocks is consistently higher than that in the complete surrounding rock. Under the 45° fractured surrounding rock, the proportion of compression-shear cracks reaches the maximum of 96.1%.

**Table 4.** Statistical results of the number of cracks.

| Fracture Angle | Total Number of Cracks | Compression-Shear Crack | | Tensile Crack | |
|---|---|---|---|---|---|
| | | Number of Cracks | Proportion | Number of Cracks | Proportion |
| 0° | 900 | 839 | 93.2% | 61 | 6.8% |
| 30° | 1013 | 952 | 94.0% | 61 | 6.0% |
| 45° | 855 | 822 | 96.1% | 33 | 3.9% |
| 60° | 1066 | 995 | 93.3% | 71 | 6.7% |
| 90° | 835 | 780 | 93.4% | 55 | 6.6% |
| Intact | 779 | 713 | 91.5% | 66 | 8.5% |

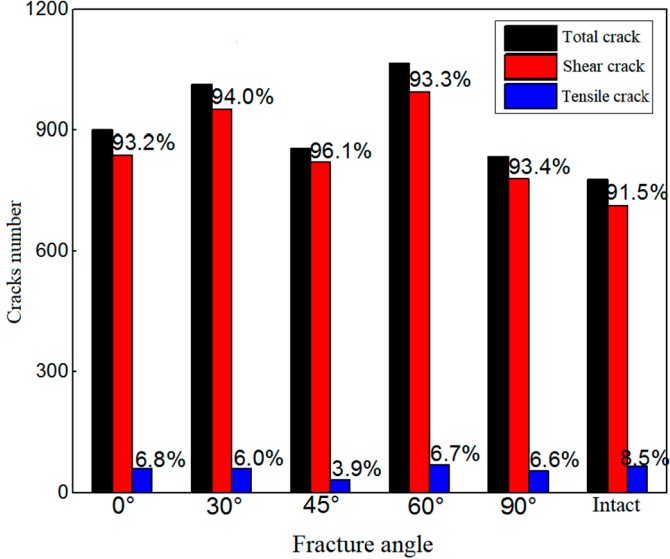

**Figure 16.** The variation diagram of fracture with fracture angle.

### 3.2.4. Energy Conversion Law

When the deep-buried tunnel is under transient unloading, the elastic strain energy stored in the surrounding rock is suddenly and rapidly released, resulting in structural damage to the surrounding rock [29–32]. The expansion of the primary fracture and the

germination of the new fracture will produce a large amount of dissipation energy. The release amount of strain energy and the total amount of dissipation energy in the system caused by transient unloading are monitored by the program, and their change rules under different fracture inclination angles are compared and analyzed as shown in Figure 17. It can be seen that the released amount of strain energy and the total amount of dissipation energy during the excavation of the surrounding rock at different fracture angles are lower than those in the integral surrounding rock. The maximum value of the released amount of strain energy in the fractured surrounding rock is 27.6% lower than that in the integral surrounding rock, and the maximum value of the total amount of dissipation energy is about 24.8% lower than that in the complete surrounding rock. The released amount of strain energy first increases slowly and then decreases rapidly with the increase in the fracture angle, and it reaches the maximum value in the 45° fractured surrounding rock. The total amount of dissipation energy increases monotonically with the increase in the fracture angle. Its variation characteristics are likely closely related to the shape of the tunnel.

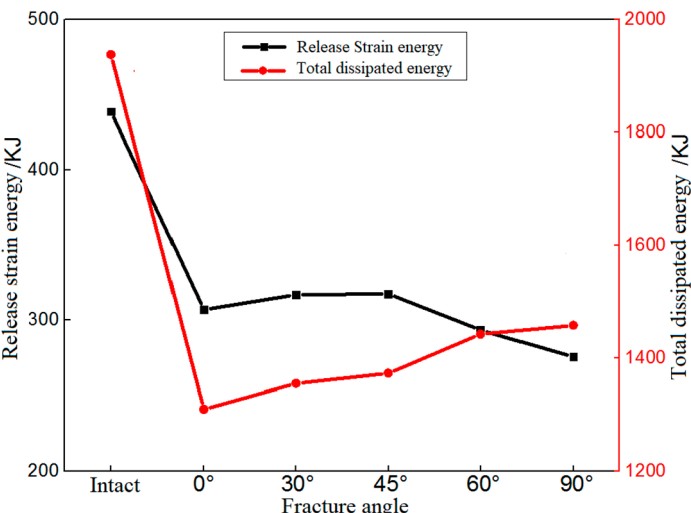

**Figure 17.** Energy comparison diagram under different fracture angles.

## 4. Conclusions

In order to explore the influence of fracture distribution on the transient unloading effect of the deep-buried tunnel, a numerical simulation was carried out for different fracture lengths and angles. Based on the analysis of stress characteristics, deformation laws, fracture propagation characteristics, and the energy conversion law of the tunnel-surrounding rock, the main conclusions are as follows:

(1) The fracture has an important impact on the stress adjustment process of transient unloading in the deep-buried tunnel. The existence of fractures forces the stress adjustment area to change after tunnel excavation. In the surrounding rock of horizontal penetrating fractures with different lengths, the displacement of the vault is always greater than that of the bottom plate, but the difference decreases with the fracture length.

(2) With the increase in the fracture length, the distance from the stress concentration at the fracture tip to the free surface gradually increases, and the damage range of the surrounding rock continues to expand. The distribution of mesocracks on the two sides of the tunnel gradually becomes sparse, the extended area gradually becomes larger, and the damaged area of the surrounding rock gradually migrates to the deep area.

(3) With the increase in the fracture length, the total number of mesocracks shows a monotonic decreasing trend. With the increase in the fracture length, the released amount of strain energy during tunnel excavation shows a trend of first decreasing

and then slowly increasing, while the total amount of dissipation energy first decreases and then increases rapidly.

(4) Under different fracture angles, the number of mesocracks varies greatly. The total number of mesocracks in the 60° and 30° fractured surrounding rocks is greater, followed by the 0° fractured surrounding rock, and the total number of mesocracks in the 45° and 90° fractured surrounding rocks is relatively less. The proportion of compression-shear cracks shows a trend of increasing first and then decreasing with the increase in the fracture angle. However, the number of mesocracks in the fractured surrounding rocks is consistently higher than that in the complete surrounding rock. Under the 45° fractured surrounding rock, the proportion of compression-shear cracks reaches the maximum value of 96.1%.

**Author Contributions:** X.L.: conceptualization, writing—original draft, and funding acquisition; G.W.: data curation, methodology, writing—review and editing, and funding acquisition; Z.W.: formal analysis, writing—review and editing, and supervision; D.W.: writing—review and editing, supervision, and funding acquisition; L.S., M.L. and H.C.: writing—review and editing, and supervision. All authors have read and agreed to the published version of the manuscript.

**Funding:** The authors acknowledge the financial support provided by the National Natural Science Foundation of China (No. 52079098), the Fundamental Research Funds for the Central Universities (No. 2042022kf1219), the Collaborative Innovation Center for Prevention and Control of Mountain Geological Hazards of Zhejiang Province (No. PCMGH-2021-03), the State Key Laboratory of Mining Disaster Prevention and Control (Shandong University of Science and Technology), Ministry of Education (No. MDPC202023), and Shaoxing Science and Technology Plan Project (No. 2022A13003).

**Institutional Review Board Statement:** Not applicable.

**Informed Consent Statement:** Not applicable.

**Data Availability Statement:** The data analyzed or generated during the research can be provided by the corresponding author upon request.

**Conflicts of Interest:** The authors declare that there are no conflict of interest.

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
