# Peer review of "The Transient Unloading Response of a Deep-Buried Single Fracture Tunnel Based on the Particle Flow Method"

_sustainability, doi:10.3390/su15086840_

Round 1

Reviewer 1 Report

The authors use the particle flow numerical simulation to reproduce the transient unloading process of deep buried single fracture tunnel. Results showed complete discussion about displacement, release strain and total dissipated energies with different fracture lengths and angles, respectively. The arguments and discussion of findings are coherent, balanced and compelling.

Reviewer 2 Report

1.  In this paper, joint and fracture are mixed to express the same meaning. For example, the joint in line 126 and the fracture in line 149.

2. The heights of tunnel side walls and arches are not indicated in Figure 2; The label in Figure 6 is not clear.

3. The fracture positions mentioned in lines 126 and 443 have not been studied as variables, but should be the key points of tunnels with different fracture lengths and angles. Please correct it.

Reviewer 4 Report

This paper uses the particle flow numerical simulation to reproduce the transient unloading process of the deeply buried single-joint tunnel. The influence of fracture characteristics on the transient unloading effect of deep rock mass is analyzed. The research is technically sound, and the conclusions are well supported by the simulations and observations. The English language might need some improvements before publication.
